# Functional Response of Four Phytoseiid Mites to Eggs and First-Instar Larvae of Western Flower Thrips, *Frankliniella occidentalis*

**DOI:** 10.3390/insects15100803

**Published:** 2024-10-14

**Authors:** Viet Ha Nguyen, Ziwei Song, Duc Tung Nguyen, Thomas Van Leeuwen, Patrick De Clercq

**Affiliations:** 1Laboratory of Agrozoology, Department of Plants and Crops, Ghent University, Coupure Links 653, B-9000 Ghent, Belgium; vietha.nguyen@ugent.be (V.H.N.); thomas.vanleeuwen@ugent.be (T.V.L.); 2Key Laboratory of Green Prevention and Control on Fruits and Vegetables in South China Ministry of Agriculture and Rural Affairs, Guangdong Provincial Key Laboratory of High Technology for Plant Protection, Plant Protection Research Institute, Guangdong Academy of Agriculture Sciences, 7 Jinying Road, Tianhe District, Guangzhou 510640, China; ziweisong@139.com; 3Entomology Department, Vietnam National University of Agriculture, Trau Quy, Gia Lam, Hanoi 131000, Vietnam; nguyenductung@vnua.edu.vn

**Keywords:** *Amblyseius largoensis*, *Amblyseius swirskii*, *Proprioseiopsis lenis*, *Paraphytoseius cracentis*, *Frankliniella occidentalis*, predation capacity

## Abstract

**Simple Summary:**

The western flower thrips, *Frankliniella occidentalis*, is a common pest in many crops worldwide. Predatory mites (Acari: Phytoseiidae) are effective predators of first-instar thrips and have been extensively used for biological control of the pest. However, the short development time of thrips larvae limits the effectiveness of thrips control. Recent studies have shown that some species of phytoseiid mites are capable of consuming thrips eggs embedded in leaf tissue, thereby extending the window for effective predation on *F. occidentalis* populations. In this study, we aimed to investigate the predatory ability of three phytoseiids native to Southeast Asia and one commercially available species on western flower thrips eggs and compare it with their predation on first instars of the pest. Both at 25 °C and 30 °C, the functional response of all the studied mites was type II to first instars of the thrips, whereas it shifted to type III when thrips eggs were provided. The consumption of first-instar thrips was consistently higher than that of thrips eggs. More thrips eggs were consumed at 30 °C than at 25 °C, whereas predation on larvae was minimally affected by temperature. Our research indicates the potential of the studied mites to contribute to the suppression of *F. occidentalis* outbreaks in Southeast Asia.

**Abstract:**

The predation capacity and functional responses of adult females of the phytoseiid mites *Amblyseius largoensis* (Muma), *Proprioseiopsis lenis* (Corpuz and Rimando), *Paraphytoseius cracentis* (Corpuz and Rimando), and *Amblyseius swirskii* (Athias-Henriot) were studied on eggs and first instars of the western flower thrips, *Frankliniella occidentalis* (Pergande), in the laboratory at 25 °C and 30 °C. At both temperatures, the functional response of all four phytoseiid mites was type II to first instars of the thrips. In contrast, when offered thrips eggs, the functional response was type III. At both temperatures tested, *A. swirskii* had the highest mean daily consumption of first-instar *F. occidentalis*, followed by *A. largoensis*, *P. cracentis*, and *P. lenis*. *Amblyseius largoensis* had the shortest handling time and the highest maximum attack rate when first-instar thrips were the prey. When fed on thrips eggs, *A. largoensis* had the highest mean daily consumption, followed by *A. swirskii*, *P. cracentis*, and *P. lenis*. On thrips eggs, *A. swirskii* showed the shortest handling time and highest maximum attack rate. Our findings indicate that all four phytoseiids had a better ability to prey on first-instar larvae of *F. occidentalis* compared to thrips eggs. At 25 and 30 °C, *A. largoensis* was the better predator on thrips larvae, whereas *A. swirskii* was superior in consuming eggs of *F. occidentalis*. *Proprioseiopsis lenis* was the inferior predator on both thrips larvae and eggs compared to the other phytoseiids tested.

## 1. Introduction

The western flower thrips, *Frankliniella occidentalis* (Pergande), is a highly destructive pest of various crops, including vegetables, fruits, and ornamentals [1,2,3]. The insect is native to western North America but has since spread worldwide, becoming a significant problem in agricultural production [4]. The pest causes both direct and indirect damage to a variety of crops. Feeding deprives the plant of nutrients and leads to spots, scars, and deformations on leaves and flowers, affecting growth and causing yield losses [5]. The western flower thrips also indirectly harms plants by transmitting plant viruses such as the tospoviruses impatiens necrotic spot virus (INSV) and tomato spotted wilt virus (TSWV) [6,7,8], further exacerbating its impact on crop yields. As the negative impacts of chemical insecticides have become evident, there is a pressing need to explore alternative approaches for controlling thrips. One such alternative method is the utilization of biological control agents.

As a developing country in Southeast Asia, Vietnam is aiming to achieve the status of a high-quality agricultural product producer and exporter [9]. In recent years, the circumstances in the region have become favorable for major outbreaks of various thrips species, including *F. occidentalis* [10]. The primary approach for handling thrips outbreaks in Vietnam is through the extensive use of insecticides. However, due to increasing concerns from both consumers and authorities in the region regarding the potential health risks associated with pesticide residue on agricultural products [11], there is a growing trend of considering biological control methods as a substitute for chemical methods [12]. An example of this trend is seen in recent research conducted in Vietnam, focusing on the potential of phytoseiid mites for the augmentative biological control of *F. occidentalis* and other arthropod pests [13].

In our previous study, the development and reproduction of three phytoseiid mites, *Amblyseius largoensis* (Muma), *Proprioseiopsis lenis* (Corpuz and Rimando), and *Amblyseius swirskii* (Athias-Henriot), were assessed on either a diet of *F. occidentalis* eggs or larvae of the thrips [14]. In the present study, we investigated the predation efficacy on eggs and first instars of *F. occidentalis* by the above phytoseiid mites as well as *Paraphytoseius cracentis* (Corpuz and Rimando). The latter species has been recorded as a thrips predator in several countries of the Asia-Pacific region, including China, Japan, New Caledonia, Papua New Guinea, the Philippines, Singapore, and Thailand [15]. It is also found in Vietnam, along with *A. largoensis* and *P. lenis*, on various crops including cucumber, pumpkin, eggplant, and chili pepper [16]. In recent surveys, *A. largoensis*, *P. lenis*, and *P. cracentis* have been commonly recorded in the Red River Delta of northern Vietnam [17]. Research on these three species has remained limited, and there are few data on their predation activity compared to commercially available species like *A. swirskii*.

The foraging behavior of predators, including their functional and numerical responses, constitutes key elements in selecting predatory mites for biological control purposes [18]. The functional response refers to the correlation between a single predator’s predation rates and varying densities of its prey over a specific period [19,20,21]. To date, most studies have focused on the predation by phytoseiids on first-instar larvae of *F. occidentalis*, whereas far fewer published studies have examined their predatory behavior toward thrips eggs [14,22,23]. Comparing predation capacity on thrips eggs and larvae helps evaluate pest control effectiveness by selecting the appropriate predator species to release against the critical developmental stages of thrips.

The present laboratory study aimed to compare the predation rates and functional responses of female adults of *A. swirskii*, *A. largoensis*, *P. cracentis*, and *P. lenis* on eggs versus first instars of *F. occidentalis*. As the study was conducted with the potential role of these phytoseiids in managing the pest in tropical Southeast Asia, where the average annual mean temperature is 25.5 °C [24], the predation experiments were carried out at temperatures of 25 and 30 °C.

## 2. Materials and Methods

### 2.1. Colonies of Thrips and Predatory Mites

Western flower thrips, *F. occidentalis*, were reared at the Department of Plants of Crops of Ghent University (Belgium) on bean pods (*Phaseolus vulgaris* Prelude) and fresh pollen of cattail, *Typha angustifolia* L. (Nutrimite, Biobest Group, Westerlo, Belgium), in plastic boxes (30 × 20 × 8 cm). Bean pods and pollen were replaced weekly. First-instar larvae and adults of *F. occidentalis* were used for the experiments.

In 2016, colonies of *P. lenis*, *A. largoensis*, and *P. cracentis* were established at Ghent University after being collected from Vietnam. *Amblyseius swirskii* was provided by BioBest Group NV (Westerlo, Belgium). To rear all predatory mites, bean leaves (*P. vulgaris*) were used as arenas. The leaves were placed upside down on a 2 cm thick layer of cotton in a plastic tray (20 × 13 × 5 cm), with a thin tissue paper layer on the leaf edges. The cotton and tissue paper were moistened to prevent the mites from escaping. First instars of *F. occidentalis* were supplied daily as the only prey. Each leaf disc was used for two weeks before the mites were transferred to a new leaf disc. All colonies of thrips and predatory mites were reared separately in Panasonic climate chambers (MLR 352H) set at 25 ± 1 °C, 65 ± 5% RH, and with a 16:8 h (L:D) photoperiod.

### 2.2. Experimental Setup

To examine the predation and functional responses of individual predatory mites, plastic dishes (5 × 1.5 cm) were employed as experimental units. The bottom was covered with a moist cotton layer (5 × 0.3 cm) upon which a section of bean leaf (diameter 4 cm) was positioned. Soft paper tissue was used to cover the leaf edges, which was moistened daily to prevent the mites and thrips larvae from escaping. Any predatory mites that did escape were excluded from data analysis. Individual deutonymphs of each phytoseiid were collected from the rearing cultures and transferred to a new leaf disc. Bean pods were placed in the rearing units of *F. occidentalis* for egg laying during 72 h. Collected pieces of bean pod containing *F. occidentalis* eggs and larvae were provided daily to deutonymphs until they completed their immature development. Newly emerged female phytoseiids were immediately paired with a male for 24 h. After the males were removed, two-day-old adult females were starved for 24 h before the start of the experiments. All experiments were conducted in a Panasonic climate chamber (MLR 352H) set at 25 ± 1 °C or 30 ± 1 °C, with a 65 ± 5% RH, and a 16:8 h (L:D) photoperiod.

#### 2.2.1. Experiments with *F. occidentalis* Larvae

Densities of 5, 10, 20, 30, 40, and 50 *F. occidentalis* first instars were offered as prey to individual predators, with 20 replicates for each prey density. Starved female phytoseiids were released into half of the dishes using a fine brush, while the other half were used as controls and contained only thrips larvae. After 24 h, the mites were removed, and the number of killed thrips larvae was calculated as the difference between the initial number and the number of survivors.

#### 2.2.2. Experiments with *F. occidentalis* Eggs

Three-day-old female thrips were allowed to lay eggs for 24 h in the bean leaf sections of the experimental arenas and then removed from the dishes. In order to observe thrips eggs on the surface of the bean leaves, a light source from below was used to illuminate the dishes under a stereomicroscope (25×). In this manner, the total number of eggs in each experimental dish could be accurately recorded. To obtain six densities of eggs, 1, 2, 5, 7, 10, and 13 female thrips were released into the experimental dishes, yielding egg numbers ranging from 4–6, 8–12, 18–22, 28–32, 38–42, and 48–52 eggs, respectively. Twenty replicates were set up for each density. Individual starved female predatory mites were introduced into 10 experimental dishes, while the remaining 10 dishes were used as controls, containing only thrips eggs. After 24 h of feeding on the eggs, the mites were removed. Seventy-two hours later, the number of hatched thrips larvae was counted. The number of *F. occidentalis* eggs consumed was calculated as the difference between the initial egg count and the number of hatched larvae.

### 2.3. Data Analysis

The analysis of the functional response data involved a two-step process, as outlined by Juliano [21]. Firstly, a logistic regression was applied to investigate the relationship between the proportion of prey consumed (*N_e_*/*N*_0_) and the initial prey density (*N*_0_), as explained in Section 2.2.1 and Section 2.2.2 above. This step allowed the identification of the shape of the functional response. Subsequently, the data were fitted to a polynomial function, enabling a description of the relationship between the proportion of prey consumed and initial density:*N_e_*/*N*_0_ = exp(P_0_ + P_1_*N*_0_ + P_2_ *N*_0_^2^ + P_3_ *N*_0_^3^)/[1 + exp(P_0_ + P_1_*N*_0_ + P_2_ *N*_0_^2^ + P_3_ *N*_0_^3^)](1)

The variable (*N_e_*/*N*_0_) represents the probability of a prey being consumed, while P_0_, P_1_, P_2_, and P_3_ denote the maximum likelihood estimates, serving as the intercept, linear, quadratic, and cubic coefficients, respectively [25]. These values were calculated using a cubic mathematical function to estimate the curve (as shown in Table 1).

To determine the type of functional response, the data were fitted to model (1). The signs of P_1_ and P_2_ were essential in distinguishing the shape of the curves. When P_1_ is negative (P_1_ < 0), it indicates a type II functional response, suggesting that the proportion of prey consumed decreases consistently with an increasing initial number of prey. On the other hand, a positive value of P_1_ and a negative value of P_2_ (P_1_ > 0 and P_2_ < 0) indicate a type III functional response, suggesting a density-dependent relationship where the proportion of prey consumed displays a more complex pattern [21].

The handling time and the attack rate coefficients for the type II functional response were determined by applying the random predator equation [26]:*N_e_* = *N*_0_{1 − exp[*a* (*T_h_ N_e_* − *T*)]}(2)
where *N_e_* is the number of prey killed; *N*_0_ is the initial number of prey; *a* is the attack rate; *T_h_* is the handling time; and *T* is the total time available for the predator (24 h).

For type III functional responses, where *a* depends on the initial prey density, the following equation is applied, where *b* is a constant [26]:*N_e_* = *N*_0_{1 − exp[*b N*_0_ (*T_h_ N_e_* − *T*)]}(3)

The data analysis was carried out using SAS software (SAS 2007). To estimate the attack rate and handling time parameters, the NLIN procedure in SAS was utilized.

Based on the estimated parameters of the functional response, the search efficiency (*E*) was determined using the equation of Beddington (1975) [27]: *E* = *a*/(1 + *a T_h_ N*_0_)(4)

Furthermore, the impact of prey densities, temperatures and prey types on the daily consumption of the phytoseiid mites was analyzed using RStudio, version 1.1.453. Shapiro-Wilk tests were used to evaluate the normality of the data. Since most of the data did not follow a normal distribution, non-parametric tests were used in this study: the data were analyzed by Mann–Whitney U tests and Kruskal-Wallis tests. *p*-values smaller than or equal to 0.05 were considered significant.

## 3. Results

At 25 °C and 30 °C, the survival rate of *A. largoensis*, *P. lenis*, *A. swirskii* and *P. cracentis* when provided with a diet of thrips eggs or thrips larvae was 100%. At both temperatures, the mortality of thrips larvae in the control was 0%, while the mortality of thrips eggs at 25 °C and 30 °C ranged from 0–2.7% and 0–3.8%, respectively. As mortality rates were lower than 5% in all control groups, predation data were not corrected [21,28].

### 3.1. Functional Response Type

At both temperatures, when thrips larvae were the prey, the number of prey consumed approached the asymptote hyperbolically as prey density increased (Figure 1A,C and Figure 2A,C), corresponding to an asymptotically declining proportion of prey killed, indicating inverse density-dependence. Logistic regression analysis examining the relationship between the proportion of thrips larvae consumed at various densities and the initial density of prey indicated a significant negative linear parameter P_1_ and a positive quadratic coefficient (Table 1 and Table 2). This suggests that the functional response of all four predatory species was type II on larvae of *F. occidentalis*. When eggs of *F. occidentalis* were used as the prey, the number of prey killed approached the asymptote as a sigmoid function and there was a rise in the proportion of prey consumed as prey density increased (Figure 1B,D and Figure 2B,D), indicating reciprocal-density dependence. Logistic regression analysis for all prey stages displayed a significant positive linear parameter (P_1_) and a negative quadratic coefficient (P_2_) (Table 1 and Table 2). This suggests that the functional response of all four predatory species was type III on eggs of *F. occidentalis*.

### 3.2. Prey Consumption, Maximum Attack Rate, and Handling Time

At 25 °C, when the diet consisted of thrips larvae, the handling times of the different phytoseiids ranged from 1.80 (*A. largoensis*) to 4.31 h (*P. lenis*). Conversely, the theoretical maximum attack rate varied from 5.53 (*P. lenis*) to 13.33 larvae per day (*A. largoensis*) (Table 3). When thrips eggs were offered, the handling times varied from 2.10 (*A. swirskii*) to 3.24 h (*P. lenis*). The maximum attack rate ranged from 7.41 (*P. lenis*) to 11.43 eggs (*A. swirskii*) (Table 3). The highest mean daily consumption of thrips larvae across all densities ranged from 13.20 larvae for *A. swirskii* to 6.20 larvae for *P. lenis* (Table 4). *Amblyseius largoensis* had the highest mean egg consumption in 24 h (4.90 eggs), compared to *A. swirskii* (4.50 eggs), *P. cracentis* (4.50 eggs), and *P. lenis* (2.40 eggs) (Table 5).

At 30 °C, when phytoseiid mites were offered larvae of *F. occidentalis*, handling times ranged from 2.04 (*A. largoensis*) to 3.98 h (*P. lenis*), with theoretical maximum attack rates of 6.03 (*P. lenis*) to 11.78 larvae (*A. largoensis*) (Table 6). When predatory mites were offered thrips eggs, the handling times ranged from 1.20 (*A. swirskii*) to 2.27 h (*P. lenis*), with maximum attack rates varying from 10.58 (*P. lenis*) to 19.96 eggs (*A. swirskii*) (Table 6). Mean daily consumption of thrips larvae recorded across all densities ranged from 12.70 larvae for *A. swirskii* to 6.10 larvae for *P. lenis* (Table 7). For thrips eggs, mean daily consumption varied between 6.40 eggs for *A. largoensis* and 3.90 eggs for *P. lenis* (Table 8). In absolute numbers, the mean daily consumption of both larvae and eggs was substantially lower for *P. lenis* compared to *A. swirskii*, *A. largoensis*, and *P. cracentis* at all prey densities, except at a density of 4–6 thrips eggs.

When phytoseiid mites were presented with larvae of *F. occidentalis*, there were no significant differences (*p* > 0.05, Mann–Whitney U tests) between 25 °C and 30 °C in the mean daily consumption rates at various densities of thrips larvae, except for *P. cracentis* at a density of 50 larvae (Appendix A). When thrips eggs were provided as prey, differences in mean daily consumption of predatory mites were observed for *A. largoensis* (at densities of 38–42 and 48–52 eggs), *P. lenis* (at densities of 28–32, 38–42, and 48–52 eggs), and *A. swirskii* (at densities of 28–32, 38–42, and 48–52 eggs) (*p* < 0.05, Mann–Whitney U tests; Appendix A). For *P. cracentis*, no differences in predation on thrips eggs were observed between 25 °C and 30 °C. At both temperatures, all predator species consumed significantly higher numbers of first instars of *F. occidentalis* compared to thrips eggs (*p* < 0.05, Mann–Whitney U tests; Appendix A).

### 3.3. Searching Efficiency

The search efficiency of the four phytoseiid mites exhibited a consistent pattern at both tested temperatures (Figure 3 and Figure 4), with values decreasing as the density of thrips larvae or eggs increased. The search efficiency of *P. lenis* on both types of prey was significantly lower compared to the other three predatory mites.

## 4. Discussion

At both 25 and 30 °C, all phytoseiid mites exhibited a type II functional response when the prey were first instars of *F. occidentalis*. Type II responses have been reported by several studies focusing on thrips predation by phytoseiid mites, including *Neoseiulus cucumeris* (Oudemans) at various densities of *F. occidentalis* first instars [29,30], *Neoseiulus barkeri* (Hughes) and *Euseius nicholsi* (Ehara & Lee) at first instars of *Thrips flavidulus* (Bagnail) [31], and *Amblyseius herbicolus* (Chant) at first instars of *Sericothrips staphylinus* Haliday [32]. A type II functional response indicates that these predatory mites increase their consumption of thrips larvae as prey availability increases at lower prey densities, and their responses gradually approach a maximum level (asymptote) at higher prey densities, albeit at a slower rate. However, when the prey were thrips eggs, the response shifted to type III, suggesting that predation was poor when egg densities were low, but predatory mites increased egg consumption with increasing availability of thrips eggs, albeit with a decreasing rate as prey density continued to rise. Likewise, some studies have reported that phytoseiids when fed on eggs as prey exhibited a type III functional response. For example, *Euseius concordis* (Chant) fed on eggs of the cassava green mite *Mononychellus tanajoa* (Bondar) [33], and *A. swirskii* fed on eggs of the two-spotted spider mite *Tetranychus urticae* (Koch) [34]. In contrast, several other studies have shown that phytoseiid mites feeding on egg prey exhibited a type II functional response, such as *Neoseiulus californicus* (McGregor) and *N. cucumeris* to eggs of *T. urticae* [35,36], *Galendromus flumenis* (Chant) to eggs of the grass mite *Oligonychus prantensis* (Banks) [37], and *A. largoensis* to eggs of the red palm mite *Raoiella indica* (Hirst) [38].

Phytoseiid mites are blind [39] and typically consume more exophytically laid eggs than other developmental stages of various arthropod prey [40,41], due to their immobility, relatively small size, and lack of defensive capabilities [42,43]. However, the eggs of *F. occidentalis* are mostly embedded within leaf tissue. Although the handling time for eggs in our study was shorter compared to that of first-instar larvae (Table 3 and Table 6), thrips eggs are still difficult for predatory mites to detect and feed upon. Indeed, the search efficiency and consumption of the tested phytoseiids for eggs of *F. occidentalis* were much lower than for first-instar larvae (Figure 3 and Figure 4, Appendix A). Mites with a high search rate, such as *A. swirskii* [43], are more likely to encounter motionless thrips eggs than “sit and wait predators”. *Amblyseius swirskii* also had the shortest handling time and the highest maximum attack rate when thrips eggs were offered compared to the other species in our study.

In Nguyen et al. [14], higher death or escape rates of *A. swirskii*, *A. largoensis*, and *P. lenis* when offered *F. occidentalis* eggs as prey during the developmental period compared to first-instar larvae suggested a preference for larvae over eggs. This preference is further corroborated by the current study, as at both temperatures and at all prey densities, the number of consumed first instars of *F. occidentalis* by all tested predatory mites was significantly higher than the number of consumed thrips eggs (Appendix A). Among the four predator species tested, *A. largoensis* had the shortest handling time and the highest maximum attack rate when first instars of *F. occidentalis* were provided. The difficulties associated with the localization and accessibility of thrips eggs compared to first-instar larvae may result in a preference for the larvae. Besides the mobility of larval prey, increasing chances of encounter, other factors may also be involved in the localization of larval thrips prey by phytoseiid mites. Previous studies have indicated that when certain arthropod prey damages a host plant, the latter may release herbivore-induced plant volatiles (HIPVs) [44,45,46,47] that are attractive to predatory arthropods, including phytoseiid mites [48,49,50]. In addition, some reports suggest that certain chemical cues released by prey when threatened can aid predatory mites in locating the prey more easily [51,52]. In the present experiments, the larvae of *F. occidentalis* caused more damage to the bean leaf tissue than the eggs did. This could be associated with a greater release of chemical cues in the presence of larval prey, leading to higher predation rates on thrips larvae than on eggs. Also, the tendency of thrips larvae to feed in groups may make them comparatively easier targets for predation [53]. Finally, first-instar larvae of *F. occidentalis* are smaller than phytoseiid adults and have weaker defenses against predator attacks than second instars and adult thrips, making the first instars relatively easy prey to subdue [54,55]. A number of other studies have reported a preference of phytoseiids for early-instar larvae of arthropod prey over eggs. For example, *Euseius alatus* (DeLeon) and *Iphiseiodes zuluagai* (Denmark & Muma) preferred larvae over eggs, nymphs, and adults of the flat mite *Brevipalpus phoenicis* (Geijskes) [56]; *A. swirskii* preferred second instars over first instars or eggs of the silverleaf whitefly *Bemisia tabaci* (Gennadius) [57]; *N. californicus* preferred larvae over nymphs and eggs of the two-spotted spider mite *T. urticae* [58]; *Neoseiulus longispinosus* (Evans) preferred larvae over eggs and adults of *Oligonychus biharensis* (Hirst) [59].

Besides prey type, several other factors may influence the functional response of predatory arthropods, including environmental temperature [35,60]. Temperature is a key factor affecting the predation behavior of predatory mites, and it may partly account for the differences observed among studies [61,62]. In the present study, temperatures within the range of 25–30 °C did not affect the type of functional response of adult females of four (sub)tropical phytoseiids to *F. occidentalis* larvae and eggs, but they did affect the functional response parameters. When the prey was thrips larvae, search efficiency, handling time, maximum attack rate, and daily prey consumption were overall similar at both tested temperatures (Table 3, Table 4, Table 6 and Table 7; Figure 3A and Figure 4A). The lack of a difference in predation characteristics on larval prey between 25 °C and 30 °C may be explained in part by increased activity of both predators and prey. Improved predation activity of the predators at the higher temperature may be counteracted by more intense defense behaviors of the thrips larvae. However, when the prey was thrips eggs, the predation performance of *A. swirskii*, *A. largoensis*, and *P. lenis* (but not *P. cracentis*) was affected by temperature, with higher values at 30 °C than at 25 °C (Table 3, Table 5, Table 6 and Table 8; Figure 3B and Figure 4B). At 30 °C, the thrips eggs developed faster and protruded more from the leaf surface compared to 25 °C, thereby facilitating prey localization and creating favorable conditions for predation, which can explain the higher egg consumption rates of *A. swirskii*, *A. largoensis*, and *P. lenis* at the former temperature.

In conclusion, the findings of the present study suggest that *A. swirskii*, *A. largoensis*, and *P. cracentis* are more effective than *P. lenis* in reducing *F. occidentalis* populations, showing a preference for first instars over eggs. An increase in temperature from 25 °C to 30 °C significantly increased mite consumption of thrips eggs, especially at higher egg densities, while their ability to prey on thrips larvae varied little with temperature. Our findings indicate the potential of the studied predatory mites to suppress *F. occidentalis* outbreaks in the tropical climate of Southeast Asia. However, further field studies are needed to fully appreciate their role in local programs for augmentative or conservation biological control, particularly considering their interactions with host plants [63] and with other predator and prey species [58], under varying climatic conditions [35].

## Figures and Tables

**Figure 1 insects-15-00803-f001:**
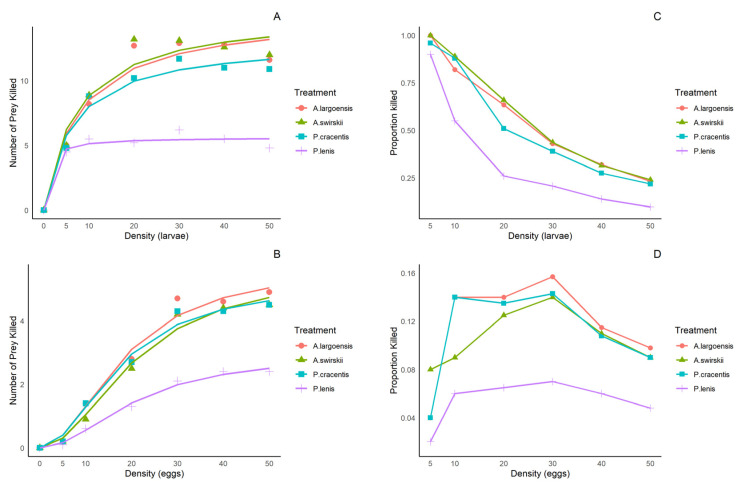
The relationships between the number of prey presented and the number of prey killed, and the corresponding relationships between the number of prey presented and the proportion killed by four phytoseiid mites when offered first-instar larvae (**A**,**C**) and eggs (**B**,**D**) of *F. occidentalis* at various densities at 25 ± 1 °C.

**Figure 2 insects-15-00803-f002:**
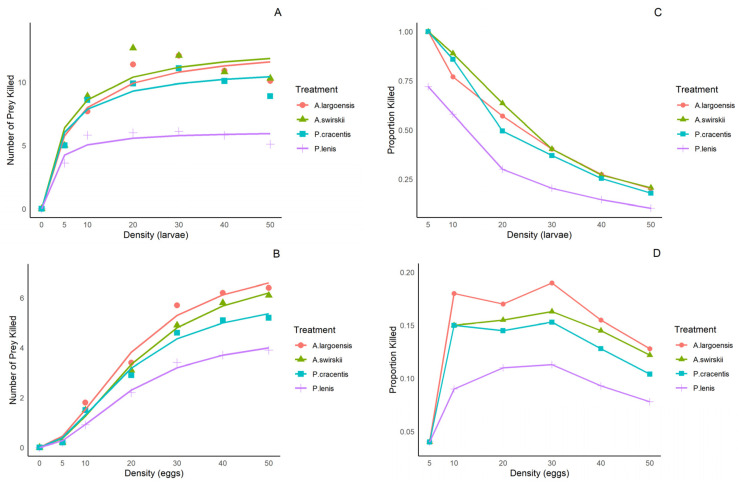
The relationships between the number of prey presented and the number of prey killed, and the corresponding relationships between the number of prey presented and the proportion killed by four phytoseiid mites when offered first-instar larvae (**A**,**C**) and eggs (**B**,**D**) of *F. occidentalis* at various densities at 30 ± 1 °C.

**Figure 3 insects-15-00803-f003:**
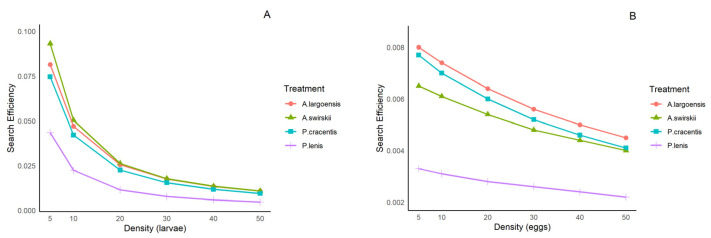
Searching efficiency of four phytoseiid mites when offered first-instar larvae (**A**) and eggs (**B**) of *F. occidentalis* at various densities at 25 ± 1 °C.

**Figure 4 insects-15-00803-f004:**
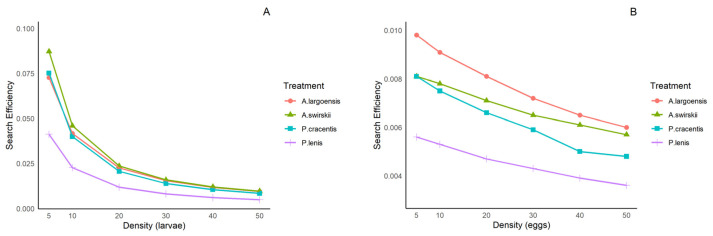
Searching efficiency of four phytoseiid mites when offered first-instar larvae (**A**) and eggs (**B**) of *F. occidentalis* at various densities at 30 ± 1 °C.

**Table 1 insects-15-00803-t001:** Maximum likelihood estimates from logistic regression of the proportion of prey consumed as a function of initial prey densities by female adults of *A. largoensis*, *P. lenis*, *A.* swirskii, and *P. cracentis* at 25 ± 1 °C.

Species	Parameters	*F. occidentalis* Larvae			*F. occidentalis* Eggs		
		Estimate	χ^2^	*p*	Estimate	χ^2^	*p*
*A. largoensis*	P_0_	4.2715 ± 0.7718	30.63	<0.0001	−2.7792 ± 0.8430	10.87	0.0010
	P_1_	−0.2971 ± 0.0875	11.52	0.0007	0.1109 ± 0.1036	1.16	0.2845
	P_2_	0.00648 ± 0.00303	4.56	0.0328	−0.00341 ± 0.00377	0.82	0.3649
	P_3_	−5 × 10^−5^ ± 3.2 × 10^−5^	2.86	0.0908	28 × 10^−6^ ± 41 × 10^−6^	0.47	0.4917
*P. lenis*	P_0_	3.4007 ± 0.5822	34.11	<0.0001	−4.2545 ± 1.3476	9.97	0.0016
	P_1_	−0.4023 ± 0.0776	26.89	<0.0001	0.1607 ± 0.1588	1.02	0.3116
	P_2_	0.0113 ± 0.00295	14.55	0.0001	−4.67 × 10^−3^ ± 0.00563	0.69	0.4073
	P_3_	−11 × 10^−5^ ± 3.3 × 10^−5^	10.88	0.0010	3.9 × 10^−5^ ± 6.1 × 10^−5^	0.42	0.5173
*A. swirskii*	P_0_	5.2190 ± 0.9361	31.09	<0.0001	−3.2677 ± 0.9578	11.64	0.0006
	P_1_	−0.3558 ± 0.1013	12.32	0.0004	0.1297 ± 0.1152	1.27	0.2601
	P_2_	0.00755 ± 0.00340	4.92	0.0266	−0.00368 ± 0.00413	0.79	0.3727
	P_3_	−6 × 10^−5^ ± 3.5 × 10^−5^	8.16	0.0920	29 × 10^−6^ ± 45 × 10^−6^	0.42	0.5180
*P. cracentis*	P_0_	4.7736 ± 0.7958	35.98	<0.0001	−2.7074 ± 0.8448	10.27	0.0014
	P_1_	−0.3852 ± 0.0897	18.43	<0.0001	0.1024 ± 0.1047	0.96	0.3280
	P_2_	0.00933 ± 0.00310	9.06	0.0026	−0.00331 ± 0.00382	0.75	0.3872
	P_3_	−8 × 10^−5^ ± 3.3 × 10^−5^	6.08	0.0137	28 × 10^−6^ ± 42 × 10^−6^	0.45	0.5001

**Table 2 insects-15-00803-t002:** Maximum likelihood estimates from logistic regression of the proportion of prey consumed as a function of initial prey densities by female adults of *A. largoensis*, *P. lenis*, *A. swirskii*, and *P. cracentis* at 30 ± 1 °C.

Species	Parameters	*F. occidentalis* Larvae			*F. occidentalis* Eggs		
		Estimate	χ^2^	*p*	Estimate	χ^2^	*p*
*A. largoensis*	P_0_	3.8361 ± 0.6977	30.23	<0.0001	−2.4671 ± 0.7656	10.39	0.0013
	P_1_	−0.2813 ± 0.0818	11.84	0.0006	0.0967 ± 0.0941	1.06	0.3040
	P_2_	0.00619 ± 0.00290	4.56	0.0327	−0.00279 ± 0.00342	0.67	0.4145
	P_3_	−5 × 10^−5^ ± 3.1 × 10^−5^	2.89	0.0893	21 × 10^−6^ ± 37 × 10^−6^	0.33	0.5652
*P. lenis*	P_0_	2.0119 ± 0.5225	14.82	<0.0001	−3.0444 ± 0.9614	10.03	0.0015
	P_1_	−0.2201 ± 0.0714	9.50	0.0021	0.0972 ± 0.1176	0.68	0.4088
	P_2_	0.00480 ± 0.00277	3.01	0.0828	−0.00286 ± 0.00426	0.45	0.5022
	P_3_	−4 × 10^−5^ ± 3.1 × 10^−5^	1.73	0.1887	23 × 10^−6^ ± 46 × 10^−6^	0.24	0.6223
*A. swirskii*	P_0_	5.2074 ± 0.9345	31.05	<0.0001	−2.5559 ± 0.8055	10.07	0.0015
	P_1_	−0.3541 ± 0.1017	12.12	0.0005	0.0874 ± 0.0987	0.78	0.3763
	P_2_	0.00716 ± 0.00343	4.36	0.0369	−0.00246 ± 0.00357	0.47	0.4916
	P_3_	−5 × 10^−5^ ± 3.6 × 10^−5^	2.28	0.1310	19 × 10^−6^ ± 39 × 10^−6^	0.24	0.6270
*P. cracentis*	P_0_	5.5762 ± 0.8561	42.43	<0.0001	−2.5366 ± 0.8142	9.71	0.0018
	P_1_	−0.4771 ± 0.0956	24.92	<0.0001	0.0844 ± 0.1006	0.70	0.4015
	P_2_	0.0124 ± 0.00328	14.22	0.0002	−0.00250 ± 0.00366	0.47	0.4946
	P_3_	−11 × 10^−5^ ± 3.3 × 10^−5^	10.73	0.0011	19 × 10^−6^ ± 40 × 10^−6^	0.23	0.6288

**Table 3 insects-15-00803-t003:** Estimates of handling time (*T_h_*) and maximum attack rate (*T*/*T_h_*) of predatory mites when fed on first-instar larvae or eggs of *F. occidentalis* at 25 ± 1 °C.

Species	*F. occidentalis* Larvae		*F. occidentalis* Eggs	
	*T_h_* (h)	*T*/*T_h_*	*T_h_* (h)	*T*/*T_h_*
*A. largoensis*	1.80 ± 0.08 (1.64–1.96)	13.33	2.15 ± 0.68 (0.79–3.51)	11.15
*P. lenis*	4.31 ± 0.19 (3.95–4.67)	5.53	3.24 ± 1.89 (−0.55–7.02)	7.41
*A. swirskii*	1.83 ± 0.07 (1.70–1.95)	13.14	2.10 ± 0.60 (0.89–3.31)	11.43
*P. cracentis*	2.07 ± 0.08 (1.91–2.22)	11.62	2.49 ± 0.60 (1.28–3.69)	8.39

Values are presented as means ± SE. The values in parentheses represent 95% confidence intervals.

**Table 4 insects-15-00803-t004:** Mean daily prey consumption by *A. largoensis*, *P. lenis*, *A. swirskii*, and *P. cracentis* at various densities of *F. occidentalis* first-instar larvae at 25 ± 1 °C.

Phytoseiid Species	*F. occidentalis* Larval Density						χ^2^	*p*
	5	10	20	30	40	50		
*A. largoensis*	5.00 ± 0.00 a A	8.20 ± 0.36 a AB	12.70 ± 0.79 a C	12.90 ± 0.67 a C	12.80 ± 0.83 a C	11.60 ± 0.58 a BC	39.970	1.51 × 10^−6^
*P. lenis*	4.50 ± 0.22 b A	5.50 ± 0.45 b AB	5.20 ± 0.49 b AB	6.20 ± 0.36 b B	5.50 ± 0.22 b AB	4.80 ± 0.44 b AB	12.341	0.030
*A. swirskii*	5.00 ± 0.00 a A	8.90 ± 0.28 a AB	13.20 ± 0.55 a C	13.10 ± 0.60 a C	12.60 ± 0.48 a C	12.00 ± 0.58 a BC	41.431	7.68 × 10^−8^
*P. cracentis*	4.80 ± 0.13 ab A	8.80 ± 0.42 a AB	10.20 ± 0.66 ab B	11.70 ± 0.58 a B	11.00 ± 0.42 a B	10.90 ± 0.66 a B	34.297	2.08 × 10^−6^
χ^2^	8.537	20.291	26.206	23.708	25.64	22.68		
*p*	0.036	1.48 × 10^−4^	8.64 × 10^−6^	2.88 × 10^−5^	1.13 × 10^−5^	4.71 × 10^−5^		

Values are presented as means ± SE. Means followed by different lowercase letters within a column are significantly different (*p* < 0.05, Kruskal–Wallis test). Means followed by different capital letters within a row are significantly different (*p* < 0.05, Kruskal–Wallis test).

**Table 5 insects-15-00803-t005:** Mean daily prey consumption by *A. largoensis*, *P. lenis*, *A. swirskii*, and *P. cracentis* at various densities of F. occidentalis eggs at 25 ± 1 °C.

Phytoseiid Species	*F. occidentalis* Egg Density						χ^2^	*p*
	4–6	8–12	18–22	28–32	38–42	48–52		
*A. largoensis*	0.20 ± 0.13 a A	1.40 ± 0.22 a A	2.80 ± 0.29 a AB	4.70 ± 0.45 a B	4.60 ± 0.48 a B	4.90 ± 0.43 a B	50.380	1.16 × 10^−9^
*P. lenis*	0.10 ± 0.10 a A	0.60 ± 0.16 b A	1.30 ± 0.30 b AB	2.10 ± 0.23 b B	2.40 ± 0.22 b B	2.40 ± 0.27 b B	46.986	5.71 × 10^−9^
*A. swirskii*	0.20 ± 0.13 a A	0.90 ± 0.23 ab A	2.50 ± 0.31 a AB	4.20 ± 0.20 a BC	4.40 ± 0.27 a BC	4.50 ± 0.17 a C	51.408	7.14 × 10^−10^
*P. cracentis*	0.20 ± 0.13 a A	1.40 ± 0.16 a A	2.70 ± 0.26 a AB	4.30 ± 0.37 a B	4.30 ± 0.33 a B	4.50 ± 0.31 a B	48.531	2.77 × 10^−9^
χ^2^	0.507	9.717	11.075	19.235	17.687	19.349		
*p*	0.918	0.021	0.011	2.45 × 10^−4^	5.10 × 10^−4^	2.32 × 10^−4^		

Values are presented as means ± SE. Means followed by different lowercase letters within a column are significantly different (*p* < 0.05, Kruskal–Wallis test). Means followed by different capital letters within a row are significantly different (*p* < 0.05, Kruskal–Wallis test).

**Table 6 insects-15-00803-t006:** Estimates of handling time (*T_h_*) and maximum attack rate (*T*/*T_h_*) of predatory mites when fed on first-instar larvae or eggs of *F. occidentalis* at 30 ± 1 °C.

Species	*F. occidentalis* Larvae		*F. occidentalis* Eggs	
	*T_h_* (h)	*T*/*T_h_*	*T_h_* (h)	*T*/*T_h_*
*A. largoensis*	2.04 ± 0.07 (1.89–2.18)	11.78	1.44 ± 0.37 (0.71–2.18)	16.64
*P. lenis*	3.98 ± 0.20 (3.58–4.38)	6.03	2.27 ± 0.77 (0.73–3.80)	10.58
*A. swirskii*	2.06 ± 0.07 (1.92–2.20)	11.65	1.20 ± 0.40 (0.39–2.01)	19.96
*P. cracentis*	2.35 ± 0.09 (2.18–2.51)	10.22	1.85 ± 0.48 (0.88–2.82)	12.98

Values are presented as means ± SE. The values in parentheses represent 95% confidence intervals.

**Table 7 insects-15-00803-t007:** Mean daily prey consumption by *A. largoensis*, *P. lenis*, *A. swirskii*, and *P. cracentis* at various densities of *F. occidentalis* first-instar larvae at 30 ± 1 °C.

Phytoseiid Species	*F. occidentalis* Larval Density						χ^2^	*p*
	5	10	20	30	40	50		
*A. largoensis*	5.00 ± 0.00 a A	7.70 ± 0.37 ab AC	11.40 ± 0.40 ab B	12.10 ± 0.57 a B	10.90 ± 0.46 a BC	10.10 ± 0.48 a BC	43.583	2.82 × 10^−8^
*P. lenis*	3.60 ± 0.40 b A	5.80 ± 0.36 b AB	6.00 ± 0.42 b B	6.10 ± 0.57 b B	5.80 ± 0.53 b AB	5.10 ± 0.35 b AB	16.299	6.04 × 10^−3^
*A. swirskii*	5.00 ± 0.00 a A	8.90 ± 0.35 a AB	12.70 ± 0.54 a C	12.10 ± 0.50 a C	10.80 ± 0.66 a C	10.30 ± 0.47 a BC	40.165	1.38 × 10^−7^
*P. cracentis*	5.00 ± 0.00 a AB	8.60 ± 0.37 a BC	9.90 ± 0.57 ab C	11.10 ± 0.53 a C	10.10 ± 0.67 a C	8.90 ± 0.38 a C	33.332	3.23 × 10^−6^
χ^2^	20.526	20.513	27.093	23.244	20.447	25.121		
*p*	1.32 × 10^−4^	1.33 × 10^−4^	5.63 × 10^−6^	3.59 × 10^−5^	1.37 × 10^−4^	1.46 × 10^−5^		

Values are presented as means ± SE. Means followed by different lowercase letters within a column are significantly different (*p* < 0.05, Kruskal–Wallis test). Means followed by different capital letters within a row are significantly different (*p* < 0.05, Kruskal–Wallis test).

**Table 8 insects-15-00803-t008:** Mean daily prey consumption by *A. largoensis*, *P. lenis*, *A. swirskii*, and *P. cracentis* at various densities of *F. occidentalis* eggs at 30 ± 1 °C.

Phytoseiid Species	*F. occidentalis* Egg Density						χ^2^	*p*
	4–6	8–12	18–22	28–32	38–42	48–52		
*A. largoensis*	0.20 ± 0.13 a A	1.80 ± 0.13 a A	3.40 ± 0.31 a AB	5.70 ± 0.37 a B	6.20 ± 0.29 a B	6.40 ± 0.34 a B	50.380	1.16 × 10^−9^
*P. lenis*	0.20 ± 0.13 a A	0.90 ± 0.23 b A	2.20 ± 0.25 a AB	3.40 ± 0.27 b B	3.70 ± 0.21 b B	3.90 ± 0.28 b B	46.986	5.71 × 10^−9^
*A. swirskii*	0.20 ± 0.13 a A	1.50 ± 0.22 ab A	3.10 ± 0.28 a AB	4.90 ± 0.23 a BC	5.80 ± 0.33 a BC	6.10 ± 0.31 a C	51.408	7.14 × 10^−10^
*P. cracentis*	0.20 ± 0.13 a A	1.50 ± 0.17 ab A	2.90 ± 0.31 a AB	4.60 ± 0.31 ab B	5.10 ± 0.35 ab B	5.20 ± 0.25 ab B	48.531	2.77 × 10^−9^
χ^2^	0	9.731	7.699	17.495	20.339	21.083		
*p*	1	0.021	0.053	5.59 × 10^−4^	1.44 × 10^−4^	1.01 × 10^−4^		

Values are presented as means ± SE. Means followed by different lowercase letters within a column are significantly different (*p* < 0.05, Kruskal–Wallis test). Means followed by different capital letters within a row are significantly different (*p* < 0.05, Kruskal–Wallis test).

## Data Availability

The data and materials generated or analyzed during this study are available upon reasonable request. Researchers interested in accessing the dataset or materials can contact the corresponding author, Patrick De Clercq, through his ORCID profile (https://orcid.org/0000-0003-0664-1602) at patrick.declercq@ugent.be.

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
