# Peer review of "Functional Response of Four Phytoseiid Mites to Eggs and First-Instar Larvae of Western Flower Thrips, *Frankliniella occidentalis"

_insects, 2024, doi:10.3390/insects15100803_

Round 1

Reviewer 1 Report

Comments and Suggestions for Authors

This study investigated the functional responses of four predatory mites on eggs and first instars of Frankliniella occidentalis. The capable consuming of predatory mites on thrips eggs embedded in leaf tissue would provide valuable information for the control of thrips, thereby extending the effective predation on F. occidentalis populations. However, I have some issues that should be clarified or addressed before the paper could be acceptable in insects.

See specific comments below.

Abstract:

Line 31: Bracket in the formal latin name is missing. Be consistent throughout the text.

Introduction:

Introduction needs strengthening in, e.g., providing stronger rationale on why authors selected these four phytoseiid mite species and compared their predation capacities on eggs versus first instars of thrips. Furthermore, it was essential to justify the choice of studying the temperatures of 25℃ and 30℃, particularly within the context of Southeast Asia. Relevant references should be added to support these decisions.

Line 51: Citation missing for this sentence.

Line 52: Please provide abbreviation about tomato spotted wilt virus.

Line 66: Replace “predatory mites of the Phytoseiidae” with “phytoseiid mites”

Line 74-75: Hong Kong and Taiwan should be included in China, and not be listed as nations like Japan and Thailand.

M&M:

Line 94: Latin scientific names are shown in italics.

Line 99-100: Delete this sentence.

Line 127: Latin scientific names are shown in italics.

Line 135-137: Is it possible to determine the specific number of eggs on each dish? Please add supplement that the observation or recording of the number of eggs laid by female thrips. How did the authors investigate the predation rates? Clearly, the authors need to explain much better how they obtained and analyzed data. 

Line 142: Was the duration of the thrips egg development (72h) sufficient at 25?

Line 151, 165, 171: The variables in the formula are indicated in italics, check the MS and revise it.

Line 163: How to estimate type III functional response parameters?

Line 173: Replace “species” with “prey types”.

Line 175-177: The reason why the authors use non-parametric tests should be noted.

Results:

Comparing the highest mean daily consumption of thrips over all densities among the predators is not meaningful enough, since these data were not subjected to analysis for differences (e.g. 214-216, 219-220).

Line 182-183: Why predation data were not corrected, and add reference here.

Figure 1-2: Please fit Type II and Type III functional response curves instead of proportion killed curves, as the data in Figure 1-2 and Table 4-5 /7-8 are repeated.

Line 212: Delete “respectively”.

Line 218: Add “(Table 3)” after “(A. swirskii)”.

Line 228: Add “(Table 6)” after “(A. swirskii)”.

Line 231-232: Please rewrite this sentence. The mean daily consumption of P. lenis, A. swirskii, A. largoensis, and P. cracentis were equal when 4-6 thrips egg were provided (Table 8).

Line 277: P-value?

Table 1-2, 4-5, 7-8: Replace x” with ´.

Figure: The quality of the figures needs much improvement.

Discussion:

This is relatively well written, where most conclusions might be OK. However, things may change after you address my previous comments.

Line 289, 290: Bracket is missing. Be consistent throughout the text.

Supplementary

Table S3-4: Replace x” with ´

Table S1-4: The statistical significance of the difference is denoted by an asterisk.

Reviewer 2 Report

Comments and Suggestions for Authors

In the reviewed MS a group of researchers from Belgium, China, and Vietnam report on their results on rearing four species of predatory phytoseids mites on eggs and immatures of a thrips F. occidentalis. They compared the functional responses of phytoseids reared on thrips eggs and immatures and concluded that all tested phytoseiid species had a better ability to prey on first-instar thrips larvae as compared with eggs. The researchers applied logistic regression and polynomial function following methodology described previously by Juliano. The statistical methods were applied correctly, the results are interpreted adequately. Some data given in Tables could be transferred to Supplement. The Discussion is very long and could be written in more condensed style. In general the MS is well written. It needs only minor linguistic corrections. The authors are requested to include several high-quality microphotographs showing the investigated phytoseids and the experimental arena, this would make the MS more attractive for readers. Some additional remarks are given below.

23 and 32,33 it is not clear what types II and III are. Abstract should be a stand-alone resume, which could be understood even without consulting the main text. Please, find a way to explain these types and make them clear to a reader of the Abstract

40,41 does this finding mean that a mixture or combination of A. largoensis and A,swirskii could be recommended for suppressing of F. occidentalis outbreaks? Please, specify this in the Abstract

48 please, explain briefly, what kind of damage to plant this does thrips cause?

68 In a previous study – in OUR previous study

71 In this present study – present is redundant

87 please consider inserting five high quality microphotographs showing the four phytoseid mites and the thrips. This is important to be sure that the phytoseids were correctly identified and this would make this MS more attractive to readers.

88,89 this information corresponds to Results and should be transferred from Intro to Results

94, 127… F. occidentalis – italic. check in the whole MS

97,98 vs 108, 109 vs 124, 125 Please, find a way to avoid repetitions. The thrips and phytoseids were reared  in the same chamber with the same adjustments.

117 please specify how exactly the deutonymphs were determained and distinguished from adults

147 please explain what is initial density, No, in your study. It is not clear from the text

156 please, explain what Type I is? This question appears because you explained Type II and II but you omitted Type I. Please find a way to explain these types briefly in the Abstract otherwise it is confusing in terms of types II,III as it is written now. Probably you could explain all three type in the Introduction, then a reader would be prepared to read about is other parts of the MS.

166-167 Please, explain which values for T, Th, a were used in your calculation. Could you give a reference to a Table here and explain this better?

178 Did you have any problems with fungi when you reared the mites and insect?

Fig.1 and 2 (Fig.3,4 – the same comment) Please make the lines bold, the indicators larger, and the colors brighter and more contrast. It is very hard to read these Figures.

Table 4. Something is wrong with the format of this Table. Please, double check it in the final pdf

Conclusion is very long. It could be two times shorter.  

Comments on the Quality of English Language

minor
